# Inhibition of Angiogenic Factor Productions by Quercetin In Vitro and In Vivo

**DOI:** 10.3390/medicines8050022

**Published:** 2021-05-12

**Authors:** Takayuki Okumo, Atsuko Furuta, Tarou Kimura, Kanako Yusa, Kazuhito Asano, Masataka Sunagawa

**Affiliations:** 1Department of Physiology, Showa University School of Medicine, Tokyo 142-8555, Japan; tokumo@med.showa-u.ac.jp (T.O.); volttarou@med.showa-u.ac.jp (T.K.); kanakotama21@gmail.com (K.Y.); suna@med.showa-u.ac.jp (M.S.); 2Department of Medical Education, Showa University School of Medicine, Tokyo 142-8555, Japan; atsuko_f@med.showa-u.ac.jp; 3Faculty of Human Sciences, University of Human Arts and Sciences, Saitama 339-8555, Japan

**Keywords:** quercetin, allergic rhinitis, angiogenic factor, mast cell, suppression, in vitro, in vivo

## Abstract

**Background:** Angiogenesis is well known to be an important event in the tissue remodeling observed in allergic diseases. Although there is much evidence that quercetin, one of the most abundant dietary flavonoids, exerts anti-allergic effects in both human and experimental animal models of allergic diseases, the action of quercetin on angiogenesis has not been defined. Therefore, in this study, we first examined the action of quercetin on the secretion of angiogenic factors from murine mast cells in vitro. We also examined the action of quercetin on angiogenic factor secretion in the murine allergic rhinitis model in vivo. **Methods:** Mast cells (1 × 105 cells/mL) sensitized with ovalbumin (OVA)-specific murine IgE were stimulated with 10.0 ng/mL OVA in the presence or the absence of quercetin for 24 h. The concentrations of angiogenic factors, vascular endothelial growth factor (VEGF), basic fibroblast growth factor (bFGF), tumor necrosis factor-α, IL-6 and IL-8 in the supernatants were examined by ELISA. BALB/c male mice immunized with OVA were challenged intranasally with OVA every other day, starting seven days after the final immunization. These mice were then orally administered quercetin once a day for five days, starting seven days after the final immunization. Clinical symptoms were assessed by counting the number of sneezes and nasal rubbing behaviors during the 10 min period just after OVA nasal provocation. The angiogenic factor concentrations in the nasal lavage fluids obtained 6 h after nasal antigenic provocation were examined by ELISA. **Results:** Quercetin significantly inhibited the production of angiogenetic factors induced by IgE-dependent mechanisms at 5.0 µM or more. Oral administration of 25.0 mg/kg quercetin into the mice also suppressed the appearance of angiogenetic factors in nasal lavage fluids, along with the attenuation of nasal symptoms. **Conclusions:** These results strongly suggest that the inhibitory action of quercetin on angiogenic factor secretion may be implicated in the therapeutic action of quercetin on allergic diseases, especially allergic rhinitis.

## 1. Introduction

Allergic rhinitis (AR) is an IgE-dependent allergic inflammatory disease in the nasal mucosa characterized by nasal itching, sneezing, watery rhinorrhea and nasal obstruction [1,2]. These nasal symptoms are accepted to be mainly mediated by several chemical mediators such as histamine, prostaglandins and leukotrienes, which are produced by activated inflammatory cells, especially mast cells, at the site of inflammation [1,2]. Furthermore, inflammatory cytokines and chemokines like IL-4, IL-13, eotaxin and RANTES are also accepted to be implicated in the development and continuation of symptoms of AR [1,2]. In addition to these responses, structural alterations in the nasal wall have been observed in AR patients. The structural alterations observed in AR patients include epithelial disruption, goblet cell hypertrophy, basement membrane thickening and collagen deposition [3,4,5]. These histological changes are called tissue remodeling, and several types of proteins, especially matrix metalloproteinases (MMPs) and tissue inhibitors of metalloproteinases (TIMPs), are reported to be essential factors for tissue remodeling [6]. It has also been reported that angiogenesis exerts essential roles in the development of tissue remodeling during allergic responses [7,8,9]. A wide variety of inducers responsible for the development of angiogenesis have been identified, such as vascular endothelial growth factor (VEGF) [9,10], basic fibroblast growth factor (bFGF) [10,11], tumor necrosis factor-α (TNF) [11], IL-6 [12] and IL-8 [11,12], which are mainly secreted by mast cells during allergic inflammation [11,12]. Mast cells isolated from human tissues are reported to spontaneously secret several types of angiogenic factors, including VEGF and IL-8, and the ability of these mast cells to secret angiogenic factors is enhanced by the cross-linking of IgE attached to the high-affinity IgE receptor FcεRI with bivalent antigen [13]. Human mast cells are also reported to secret angiogenic factors in the absence of degranulation via the activation of the prostaglandin EP2 receptor by PGE2 stimulation [14]. Furthermore, human mast cells express the high-affinity urokinase plasminogen activator receptors, which are involved in tissue remodeling and angiogenesis [15]. In addition to the mast cells function of secreting chemical mediators (e.g., histamine and leukotrienes) and inflammatory cytokines, these reports indicate that mast cells may exert essential roles in the development of allergic inflammatory angiogenesis as final effector cells.

Recently, a number of plant extracts were reported to have significant advantages in the prevention and treatment of allergic disorders, such as AR and atopic dermatitis. Curcumin, one of the polyphenols in plants, had been reported to inhibit the production of inflammatory cytokines, IL-4, IL-8 and TNF by mononuclear cells, resulting in the improvement of nasal symptoms (e.g., sneezing and rhinorrhea) and nasal airflow in AR patients [16]. Flavonoids, such as apigenin and luteolin, were reported to attenuate the development of clinical symptoms of AR via the modification of the Th1/Th2 cytokine balance in an experimental AR model mouse [17,18]. A large number of studies have proved that quercetin, one of the most abundant natural flavonoids, could prevent the development of AR through the suppression of eosinophil activation and inflammatory cytokine production [19,20]. It was also reported that quercetin is able to inhibit nitric oxide, an important final effector molecule in the development of AR, production by nasal epithelial cells, which was increased by IL-4 stimulation in vitro [21]. Furthermore, quercetin has been reported to suppress the production of neuropeptides, such as substance P and nerve growth factor, which are able to induce the vasodilation, edema and activation of inflammatory cells in the nasal wall after antigenic stimulation in AR model rats [22]. Moreover, quercetin increased the ability of nasal epithelial cells to secret thioredoxin after antigenic stimulation in vitro and in vivo [23]. In regard to quercetin’s effect on mast cell activation, including degranulation and cytokine secretion, quercetin is reported to inhibit the secretion of chemical mediators such as histamine and leukotrienes from human and murine mast cells after IgE receptor stimulation in vitro [24,25]. Quercetin also inhibits the secretion of inflammatory cytokines such as IL-6, IL-8 and TNF from murine and human cultured mast cells after IgE stimulation in vitro [26,27]. This inhibitory action of quercetin was also observed when these mast cells were cultured with phorbol 12-myristate 13-acetate and calcium inonophore A23187 [27,28]. Since these cytokines are widely accepted to induce angiogenesis [12], these reports may suggest that the suppressive activity of quercetin on these angiogenic factor productions is partially responsible for attenuation of the clinical conditions of AR. However, the influence of quercetin on the secretion of angiogenic factors, including VEGF and basic fibroblast growth factor (bFGF), is not fully understood. Therefore, the present experiments were designed to examine the action of quercetin on angiogenic factor production in vitro and in vivo.

## 2. Materials and Methods

### 2.1. Mice

Specific pathogen-free BALB/c male mice that were 5 weeks of age were obtained from CLEA JAPAN Co., Ltd. (Tokyo, Japan). They were maintained in our animal facilities under standard conditions; that is, they were maintained under a temperature of 25 ± 2 °C, humidity of 55 ± 10% and a 12-h light/dark cycle. All animal experimental protocols were approved by the Animal Experimental Ethics Committee of Showa University (Approved number: 06010; Date: 1 April 2019).

### 2.2. Chemicals

Ovalbumin (OVA), bovine serum albumin (BSA), an RPMI-1640 medium and quercetin were obtained from SIGMA-ALDRICH Co., Ltd. (St Louise, MO, USA). Fetal bovine serum (FBS) was obtained from the Nippon Bio-Supply Center (Tokyo, Japan). Quercetin was first dissolved in dimethyl sulfoxide (DMSO) at a concentration of 100.0 mM and diluted with an RPMI-1640 medium supplemented with 10% FBS (RPMI-FBS), so as to give appropriate concentrations for the experiments. These solutions were then filtered through 0.2 µm filters and stored at 4 °C until use. Percoll was purchased from Pharmacia (Uppsala, Sweden), diluted with sterile normal saline at 72% and stored at 4 °C until use.

### 2.3. Preparation of Mouse IgE

Mice were injected intraperitoneally with 0.5 mg OVA absorbed with 5.0 mg aluminum hydroxide (alum) in 0.5 mL of phosphate buffered saline (PBS) two times every two weeks. After one week, the OVA-specific serum IgE was obtained from the blood using KAPTIVE-AE kits (TECHNOGEN, Piana di Monte Verna, Italy). After measuring the protein concentration in the IgE solution by a Bio-Rad protein assay kit (Bio-Rad Co., Ltd., Hercules, CA, USA), the solution was adjusted to 1.0 mg/mL with RPMI-FBS. This solution was then sterilized using a 0.2 µm filter and stored at −40 °C until use.

### 2.4. Preparation of Mouse Peritoneal Mast Cells

Mouse peritoneal mast cells were obtained by the method described previously [13]. Briefly, mice killed by intravenous injection with 25 mg/kg pentobarbital received intraperitoneal injection with 10 mL PBS, massaged for 2 min and PBS was collected from their peritoneal cavities. The fluids were then centrifuged at 1000× *g* for 15 min at 25 ± 2 °C. The cells were suspended in PBS that contained 1% BSA overlayered onto a 72% Percoll solution and centrifuged at 3000× *g* for 30 min at 25 ± 2 °C. The cells were obtained and resuspended in RPMI-FBS. The purity of the mast cells was 98%, as judged by alcian blue staining.

### 2.5. Sensitization of Mast Cells with IgE and Treatment with Quercetin

The mast cells (1 × 10^6^ cells/mL) were suspended in RPMI-FBS that contained 500.0 ng/mL OVA-specific IgE and cultured at 37 °C for 30 min. The cells were then washed 3 times with RPMI-FBS, suspended in the medium at 1 × 10^5^ cells/mL and used as IgE-sensitized mast cells. The sensitized cells (1.0 mL) were introduced into each well of the 24-well culture plates in triplicate, which contained 10.0 ng/mL OVA and various concentrations of quercetin in a final volume of 2.0 mL. After 24 h, supernatants were collected and stored at −40 °C until use. Cells used for examining angiogenic factor mRNA expression were also cultured in a similar manner for 12 h. In all cases, treatment of the cells with quercetin was started 2 h before OVA stimulation.

### 2.6. OVA Sensitization and Quercetin Treatment

The mice were immunized by an intraperitoneal injection of 20.0 mg/mL OVA mixed with 1.0 mg of alum in a volume of 200.0 µL on days 0, 7 and 14 [23]. On days 21, 23 and 25, the mice were intranasally instilled 100 mg of OVA (5.0 µL in PBS) [23]. The mice were administered various doses (10, 20, 25 and 30 mg/kg/0.5 mL) of quercetin via a stomach tube once a day for five consecutive days, starting on day 21 relative to the immunization.

### 2.7. Collection of Nasal Lavage Fluids

The nasal lavage fluids were obtained according to the methods described previously [23]. Briefly, the mice were intraperitoneally injected with sodium pentobarbital (50.0 mg/kg) 6 h after the OVA nasal challenge. After hair removal with a razor, the neck of the mouse was incised to expose the trachea, and the trachea was amputated. A catheter with a diameter of 1.0 mm was inserted approximately 1.0 cm from the cut trachea toward the nasal cavity, and then 1.0 mL of PBS was injected to wash the nasal cavity. The nasal outflow of liquid was collected and centrifuged at 3000 rpm for 15 min at 4 °C. After measuring the IgA concentrations with ELISA (Bethyl Lab., Inc., Montgomery, TX, USA), the fluids were used as nasal lavage fluids.

### 2.8. Assay for Nasal Symptoms

Nasal allergy-like symptoms were inspected by counting the number of sneezes and nasal rubs for 10 min just after the OVA nasal challenge. After the nasal challenge with 0.1% OVA solution in PBS (5.0 µL), the mice were placed into plastic animal cages (35 × 20 × 30 cm), and the number of sneezes and nasal rubbing movements was counted for 10 min [23].

### 2.9. Assay for Angiogenic Factor Concentrations in Culture Supernatants

The angiogenic factor, VEGF, bFGF, TNF, IL-6, and IL-8 concentrations were examined in triplicate using ELISA test kits (R & D Corp., Minneapolis, MN, USA) according to the manufacturer’s recommendations. The minimum detectable levels of these kits were 0.7 pg/mL for VEGF, 3.6 pg/mL for bFGF, 3.0 pg/mL for TNF, 1.8 pg/mL for IL-6 and 2.0 pg/mL for IL-8.

### 2.10. Assay for mRNA Expression

The mRNA expression for VEGF, bFGF, TNF, IL-6 and IL-8 was examined by real-time RT-PCR. Poly A^+^ mRNA was isolated from 12 h-cultured cells using oligo (dT)-coated magnetic micro beads (Milteny Biotec). First-strand cDNA was synthesized from 1.0 mg of Poly A^+^ mRNA using a Superscript cDNA synthesis kit (Invitrogen Corp., Carlsbad, CA, USA) following the manufacturer’s instructions. A polymerase chain reaction (PCR) was carried out using the GeneAmp 5700 Sequence Detection System (Applied Biosystems, Forster City, CA, USA). The PCR mixture included 2.0 µL of the sample cDNA solution (100.0 ng/µL), 25.0 µL of SYBR-Green Mastermix (Applied Biosystems), 0.3 µL of both sense and antisense primers and distilled water, with a total volume of 50.0 µL. The reaction was conducted as follows: 4 min at 94 °C, followed by 40 cycles for 4 min at 95 °C, 1 min at 60 °C and 1 min at 70 °C [29]. β-actin was amplified as an internal control. The angiogenic factor mRNA levels were calculated by using the comparative parameter threshold cycle and were normalized to β-actin levels. The nucleotide sequences of the primers are shown in Table 1 [29,30,31].

### 2.11. Statistical Analysis

Statistically significant differences were analyzed using ANOVA, followed by a Bonferroni multiple comparison test. Differences with *p* values < 0.05 were considered statistically significant.

## 3. Results

### 3.1. Suppression of Angiogenic Factor Secretion from Mast Cells by Quercetin

The first experiments were designed to examine the action of quercetin on the secretion of angiogenic factors, VEGF, bFGF, TNF, IL-6 and IL-8 from mast cells after antigenic stimulation. IgE-sensitized mast cells were stimulated with 10.0 ng/mL OVA in the presence of 1.0–10.0 µM quercetin for 24 h. The concentrations of angiogenic factors in the supernatants were examined by ELISA. As shown in Figure 1A, quercetin exerted suppressive effects on the mast cells to secret VEGF, which was increased by OVA stimulation. The minimum concentration that caused significant inhibition of VEGF secretion was 5.0 µM. We then examined the action of quercetin on bFGF secretion from the mast cells. Quercetin could also inhibit bFGF secretion from the mast cells, induced by OVA stimulation, when the cells were cultured with quercetin at 5.0 µM or more, but not less than 2.5 µM (Figure 1B). Furthermore, the data in Figure 1 shows the inhibitory effects of quercetin on the secretion of TNF (C), IL-6 (D) and IL-8 (E), which were increased by OVA stimulation.

### 3.2. Suppression of mRNA Expression for Angiogenic Factors by Quercetin in Mast Cells

The second experiments were designed to examine the action of quercetin on mRNA expression for angiogenic factors in mast cells induced by antigenic stimulation. The mast cells sensitized with IgE were stimulated with 10.0 ng/mL OVA in the presence of 2.5–7.5 µM quercetin for 12 h. The mRNA expression was examined by quantitative real-time RT-PCR. Treatment of the IgE-sensitized mast cells with quercetin at more than 5.0 µM, but not 2.5 µM, significantly suppressed mRNA expression for VEGF, bFGF, TNF, IL-6 and IL-8, which had been increased by OVA stimulation (Figure 2A–E).

### 3.3. Suppression of the Appearance of Angiogenic Factors in Nasal Lavage Fluids by Quercetin

The third experiments were undertaken to examine the action of quercetin on the appearance of angiogenic factors in the nasal lavage fluids obtained from the mice sensitized with OVA after nasal antigenic provocation. The mice were orally administered 10.0–30.0 mg/kg of quercetin at days 21–25 after sensitization. Nasal lavage fluids were obtained 6 h after the final nasal OVA provocation. As shown in Figure 3A, the oral administration of 25.0 and 30.0 mg/kg of quercetin, but not 10.0 and 20.0 mg/kg, could suppress VEGF concentrations in nasal lavage fluids as compared with the controls. The data in Figure 3 also show the inhibitory effect of quercetin at more than 25.0 mg/kg on the appearance of angiogenic factors, bFGF, TNF, IL-6 and IL-8 in nasal lavage fluids, which were increased by the OVA nasal challenge (Figure 3B–E).

### 3.4. Suppression of the Development of OVA-Induced Nasal Allergy-Like Symptoms

The fourth experiments were designed to examine the action of quercetin on the development of nasal allergy-like symptoms after nasal antigenic provocation. The mice sensitized with OVA were orally administered 10.0–30.0 mg/kg of quercetin at days 21–25 after sensitization. The frequency of sneezing and nasal rubs that occurred within 10 min just after nasal antigenic provocation was observed and recorded. Although lower doses (10.0 mg/kg and 20.0 mg/kg) of quercetin scarcely affected the development of nasal symptoms induced by intranasal provocation of OVA, higher doses (more than 20.0 mg/kg) of quercetin inhibited the development of symptoms, and the number of sneezes and nasal rubs of these mice was significantly lower than those observed in other groups (Figure 4).

## 4. Discussion

The results obtained in this in vitro study clearly showed that quercetin could inhibit angiogenic factor secretion from mast cells induced by antigenic stimulation via the suppression of mRNA expression for angiogenic factors. The minimum concentration that caused significant suppression was 5.0 µM. It is reported that for the oral administration of 1200 mg quercetin, which is the recommended dose as a dietary supplement to healthy humans, blood concentrations of quercetin gradually increased, peaked at 12 µM and decreased to half the peak levels by 24 h after administration [32], which were higher levels showing suppressive effects on angiogenic factor secretion in vitro. These observations, therefore, strongly suggest that the present results may reflect the in vivo effect of quercetin on angiogenic factor secretion from mast cells after antigenic stimulation. This may be supported by the observation that oral administration of quercetin at more than 25.0 mg/kg into OVA-sensitized mice was able to decrease the angiogenic factor concentrations in the nasal lavage fluids, which were increased by the intranasal provocation of a specific antigen.

Histological observations of nasal mucosa from AR patients clearly showed the increased thickness, epithelial cell detachment and fibrosis of the basement membrane and intense collagen deposition in the superficial and submucosal layers in the turbinate [3,4,5]. A significant increase in size and density of the submucosal glands and goblet cells was also reported in AR patients compared with healthy individuals [3]. Furthermore, it was reported that there was a significant increase in the density of the nerve fibers in the epithelium and around the glands in AR patients [33]. These histological changes are called tissue remodeling and are characterized by angiogenesis in the nasal mucosa [7,8,9]. Angiogenesis requires the destruction of the basement membrane by MMPs and transformation of the endothelial cells to form tubes [34]. These processes are mediated by a number of factors, such as VEGF, angiogenin, transforming growth factor and bFGF, among others [8,9,10,11,12]. Among them, VEGF and bFGF are the most important factors for angiogenesis and increase the proliferation, migration and tube formation of endothelial cells [35]. VEGF increases the secretion of both MMPs and several types of chemokine, as well as the expression of intracellular adhesion molecules and E-selectin, which are essential for inflammatory cell infiltration [36]. In regard to the development of AR, VEGF and bFGF are reported to increase vasodilation and vascular permeability, which is implicated in swelling of the nasal mucosa, the increase in watery rhinorrhea and infiltration of the inflammatory cells in the nasal walls [37]. TNF is widely accepted to be produced from inflammatory cells, including mast cells and macrophages after antigenic stimulation, and to play essential roles in the development of allergic inflammatory diseases, such as AR through the enhancement of Th2-type cytokine production and the infiltration of Th2-type T cells into the site of allergic inflammation [38]. It was also reported that TNF enhances the effect of IL-4 on antigen-specific IgE production from B cells [38]. Moreover, TNF increases the expression of endothelial leukocyte adhesion molecule-1, vascular cell adhesion molecule-1 and selectin, which are essential factors for inflammatory cell migration into the site of allergic reactions in nasal mucosa after OVA stimulation in mice [38,39]. IL-6 and IL-8 are key pro-inflammatory cytokines, which exit at the local of inflammatory sites. In addition to the function of these cytokines as angiogenic factors [40], IL-6 is an important cytokine responsible for the induction of IgE production from B cells, as well as mast cell proliferation and maturation [41]. IL-8 is reported to induce inflammatory migration and the production of chemical mediators such as leukotriene and histamine from mast cells, which are responsible for the induction of AR clinical symptoms [41]. From these reports, although the present in vitro data strongly suggest that the inhibitory action of quercetin on IgE-dependent angiogenic factor secretions from mast cells may contribute to the therapeutic action of quercetin on AR, it is necessary to examine the action of quercetin on angiogenic factor secretions in vivo. The second set of experiments, therefore, was carried out to examine whether quercetin could also suppress the angiogenic factor secretions in vivo and whether this suppressive activity was related to the development of nasal allergy-like symptoms using the experimental mouse model of AR. The data obtained from the in vivo experiments clearly show that nasal lavage fluids obtained from sensitized, non-treated mice contained higher levels of angiogenic factors compared with the control mice. It was also demonstrated that oral administration of quercetin caused a decrease in the angiogenic factor concentrations in the nasal lavage fluids, along with the attenuation of the development of nasal allergic symptoms induced by OVA nasal provocation. The minimum dose that caused significant inhibition of these parameters was 25.0 mg/kg, which is similar to the maximum daily recommended dosage of quercetin as a supplement [32]. The present in vivo results, therefore, are reasonably interpreted as the suppressive ability of quercetin on angiogenic factor secretion providing possible mechanisms that could explain the attenuating effects of quercetin on the development of AR.

Although the data obtained in this study clearly show that quercetin inhibited angiogenic factor secretion in vitro and in vivo, the precise mechanisms by which quercetin could inhibit angiogenic factor secretions are not clear at present. Our previous works clearly showed that quercetin could inhibit the activation of transcription factors such as NF-κB [20]. There is also much evidence that quercetin could inhibit the activation of MAPKs, including p38MAPK and ERK1/2 in vitro and in vivo [27]. Furthermore, it has been reported that the production of inflammatory cytokines and chemokines from mast cells after antigenic stimulation requires the activation of the MAPKs/NF-κB signaling pathway [42,43], suggesting that quercetin inhibits the MAPKs/NF-kB signaling pathway and results in the suppression of angiogenic factor secretion in vitro and in vivo. Further experiments are required to clarify this point.

## 5. Conclusions

The data obtained in the present study suggest that the suppressive activity of quercetin on angiogenic factor secretion may consist, in part, of a therapeutic action of quercetin on AR.

## Figures and Tables

**Figure 1 medicines-08-00022-f001:**
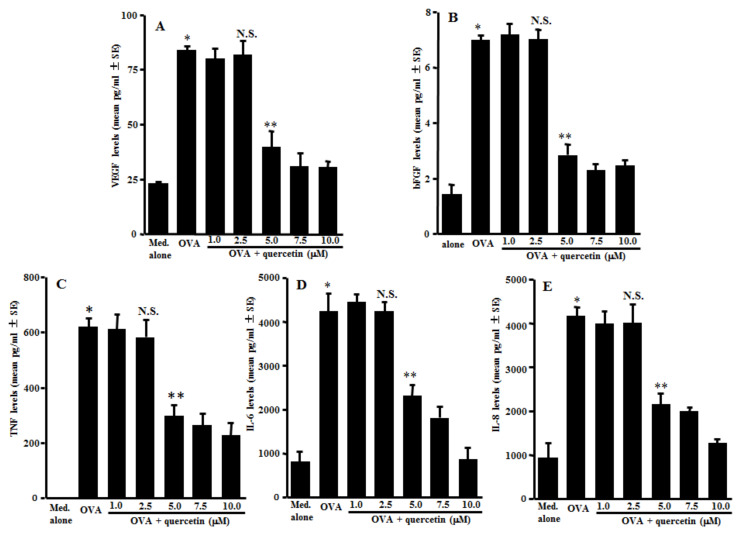
Suppression of angiogenic factor secretion from mast cells by quercetin in vitro. Mast cells (1 × 10^5^ cells/mL) sensitized with IgE were stimulated with 10.0 ng/mL of a specific antigen in the presence of quercetin for 24 h. The concentrations of VEGF (**A**), bFGF (**B**), TNF (**C**), IL-6 (**D**) and IL-8 (**E**) in culture supernatants were examined by ELISA. The data were expressed as the mean pg/mL ± SE of triplicated cultures. One representative experiment of two is shown. VEGF: vascular endothelial growth factor; bFGF: basic fibroblast growth factor; TNF: tumor necrosis factor-α; OVA: ovalbumin; and N.S.: not significant (*p* > 0.05) vs. OVA. * Significant (*p* < 0.05) vs. med. alone. ** Significant (*p* < 0.05) vs. OVA and quercetin (2.5 µM).

**Figure 2 medicines-08-00022-f002:**
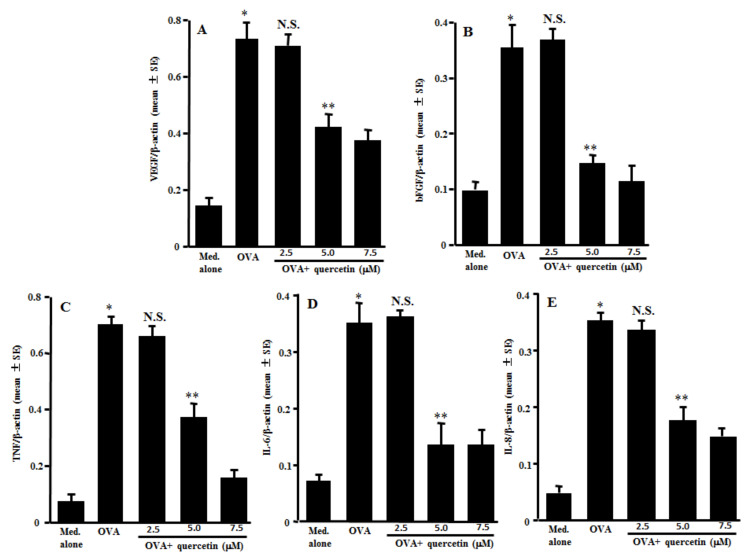
Influence of quercetin on mRNA expression for angiogenic factors in mast cells in vitro. Mast cells sensitized with IgE (1 × 10^5^ cells/mL) were stimulated with 10.0 ng/mL specific antigens in the presence of various concentrations of quercetin for 12 h. The levels of mRNA expression for VEGF (**A**), bFGF (**B**), TNF (**C**), IL-6 (**D**) and IL-8 (**E**) in mast cells were examined by real-time RT-PCR. One representative experiment of two is shown. VEGF: vascular endothelial growth factor; bFGF: basic fibroblast growth factor; TNF: tumor necrosis factor-α; OVA: ovalbumin; and N.S.: not significant (*p* > 0.05) vs. OVA. * Significant (*p* < 0.05) vs. med. alone. ** Significant (*p* < 0.05) vs. OVA and quercetin (2.5 µM).

**Figure 3 medicines-08-00022-f003:**
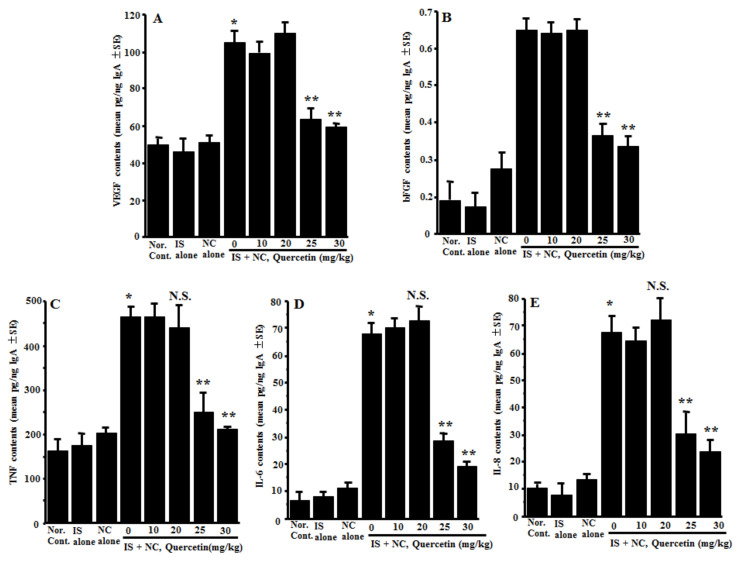
Influence of quercetin on the appearance of angiogenic factors in nasal lavage fluids obtained from OVA-sensitized mice after nasal antigenic provocation. OVA-sensitized mice were treated orally with various doses of quercetin once a day for five days before OVA nasal provocation. Nasal lavage fluids were obtained from the mice 6 h after nasal antigenic provocation. The concentration of VEGF (**A**), bFGF (**B**), TNF (**C**), IL-6 (**D**) and IL-8 (**E**) were examined by ELISA. The data are expressed as the mean pg/ng IgA ± SE of five mice. OVA: ovalbumin; VEGF: vascular endothelial growth factor; bFGF: basic fibroblast growth factor; TNF: tumor necrosis factor-α; Nor. Cont.: normal control; IS: intraperitoneal sensitization alone; NC: nasal challenge alone; and N.S.: not significant (*p* > 0.05) vs. IS + NC, quercetin (0 mg/kg). * Significant (*p* < 0.05) vs. Nor. C.; ** Significant (*p* < 0.05) vs. IS + NC, quercetin (0 mg/kg).

**Figure 4 medicines-08-00022-f004:**
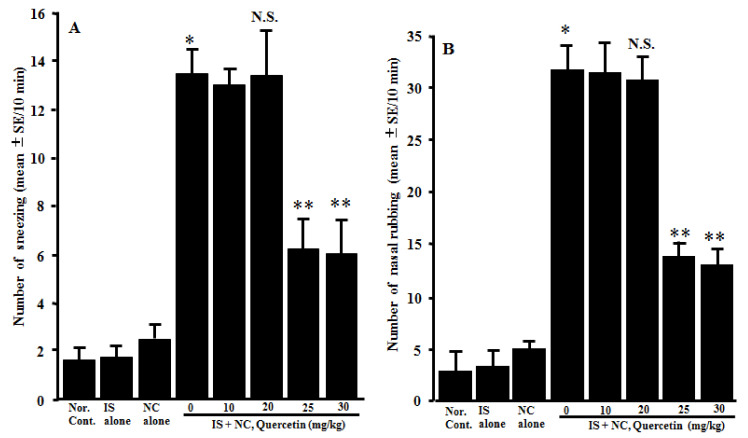
Influence of quercetin on the induction of nasal allergy-like symptoms in OVA-sensitized mice after the nasal antigenic challenge. OVA-sensitized BALB/c mice were treated orally with various doses of quercetin once a day for five consecutive days before OVA nasal provocation. Nasal allergy-like symptoms, namely the number of sneezes (**A**) and nasal rubbing movements (**B**), were counted for 10 min just after nasal antigenic provocation. The data are expressed as the mean ± SE of five mice. OVA: ovalbumin; Nor. Cont.: normal control; IS: intraperitoneal sensitization alone; NC: nasal challenge alone; and N.S.: not significant (*p* > 0.05) vs. IS + NC, quercetin (0 mg/kg). * Significant (*p* < 0.05) vs. Nor. C. ** Significant (*p* < 0.05) vs. IS + NC, quercetin (0 mg/kg).

**Table 1 medicines-08-00022-t001:** The nucleotide sequences of the primers.

Substance	Sequence	Reference
IL-6	5’-TGTGCAATGGCAATTCTGAT-3’(sense)	[30]
	5’-GGTACTCCAGAAGACCAGAGGA-3’(antisense)	
IL-8	5’-GCGCCTATCGCCAATGAG-3’(sense)	[29]
	5’-AGGGCAACACCTTCAAGCTCT-3’(antisense)	
bFGF	5’-AAGAGCGATCCGCACACTAA-3’(sense)	[31]
	5’-GGATAGCTTTCTGTCCAGGT-3’(antisense)	
VEGF	5’-CAGCTATTGCCGTCCGATTGAGA-3’(sense)	[29]
	5’-TGCTGGCTTTGGGAGGTTTGAT-3’(antisense)	
TNF	5’-CCTGTAGCCCACGTCGCGTAGC-3’(sense)	[29]
	5’-TTGACCTCAGCGCTGAGTTG-3’(antisense)	
β-actin	5’-ACCCACACTTGTGCCCATCCTA-3’(sense)	[29]
	5’-CGGAACCGCTCATTGCC-3’(antisense)	

VEGF: vascular endothelial growth factor; bFGF: basic fibroblast growth factor; TNF: tumor necrosis factor-α.

## Data Availability

Not applicable.

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
