# Peer review of "Inhibition of Angiogenic Factor Productions by Quercetin In Vitro and In Vivo"

_medicines, 2021, doi:10.3390/medicines8050022_

Round 1

Reviewer 1 Report

1.The author did not explain why MAPKs and transcription factors are related to inflammatory factors.

2. In addition, the author has not been proved that quercetin inhibits the secretion of angiogenic factors by mast cells through MAPKs/NF-kB and GATA-1. Please provide relevant supporting evidence or experimental data.

3.The results only showed that quercetin had an inhibitory effect on VEGF, TNF- and KC levels. In addition, IL-8 and other angiogenesis factors also play an important role in allergic diseases. Therefore, does quercetin also have an inhibitory effect on them so that they can conclude?

4. Line104, please correct the word "-4OoC"

Author Response

Referee 1

I greatly appreciate your valuable comments. My replies to your specific comments are as follows. Revised portions are marked with underlines.

Comment 1 & 2: The main purpose of the present study is to examine the influence of quercetin on the production of angiogenic factors from mast cells but not analyze the molecular suppressive mechanisms of quercetin on angiogenic factor production. I performed additional experiments to examine whether quercetin also exerts the suppressive effects on angiogenic factor production in vivo using experimental mouse model of allergic rhinitis. I revised the section of Materials and Methods(insert section 2.6. to 2.8.), Results(insert section 3.3. and 3.4. and Discussion(insert lines 294 to 308), which are indicated with underline. I created new 2 figures (Figure 3 and Figure 4) by using new data. The molecular mechanisms of inhibitory action of quercetin on angioggenic factor production were speculated in the last paragraph of discussion section (Lines 309 – 317) by using references. To do this, new 2 references were added (No. 40 and No. 41).

Comment 3: According to this comment, I examined protein levels of basic fibroblast growth factor and IL-6 by ELISA. I also examined mRNA expression for these angiogenic factors by RT-PCR. The new data were inserted in Figure 1 and 2.  

Other major change

           According to the comment from Referee 2, I revised the introduction section (Lines 81 – 93). To do this, new 9 references were added (No. 24 - 32).

Reviewer 2 Report

In this manuscript, Okumo et al describe the inhibitory effects of quercetin on mast cell activation. Mouse peritoneal mast cells were activated by IgE crosslinking and effects of quercetin on mediator release (TNF, VEGF, and KC) and signal transduction were measured. 

Major concerns:

  1. The inhibitory effects of quercetin on mast cell activation and mediator release in murine, rat and human mast cells have been reported in numerous publications (e.g. PMID: 22470478; PMID: 20383790; PMID: 23313938; PMID: 27498772). None of these studies were cited in the present manuscript. The current studies described in this manuscript therefore lack sufficient novel findings to warrant publication in Medicines.
  2. The authors go a long way in describing the role of angiogenic factors in allergic rhinitis. However current study does not provide any experimental evidence that inhibition of secretion of mast cell-derived angiogenic factors may contribute to the improvement of allergic rhinitis/allergic diseases. 

Author Response

Referee 2

I greatly appreciate your valuable comments. My replies to your specific comments are as follows. Revised portions are marked with underlines.

Comment 1: According to this comment, I revised the introduction section (Lines 81 – 93). To do this, new 9 references were added (No. 24 - 32).

Comment 2: According to this comment, I performed additional experiments to examine whether quercetin also exerts the suppressive effects on angiogenic factor production in vivo using experimental mouse model of allergic rhinitis. I revised the section of Materials and Methods (insert section 2.6. to 2.8.), Results (insert section 3.3. and 3.4.) and Discussion (insert lines 294 to 308), which are indicated with underline. I created new figures (Figure 3 and Figure 4) by using new data.

Other Major Change

According to the comments raised by Referee 1, I speculated the molecular mechanisms of inhibitory action of quercetin on angiogenic factor production in the last paragraph of discussion section (Lines 309 – 317) by using references. To do this, I added new 2 references (No. 40 and No. 41).

Reviewer 3 Report

The experimental design is scientifically sound and the results are interpreted and discussed appropriately. The authors should indicate how many times/biological replicates each experiment was performed. 

Author Response

Referee 3

I greatly appreciate your valuable comments. My replies to your specific comments are as follows. Revised portions are marked with underlines.

Comment 1: I performed in vitro experiments 2 times. This was inserted figure legends.

Other major changes

According to the comments raised by Referees 1 and 2, I revised the manuscript. The main portions of revision were as follows. 1) I performed additional experiments to examine whether quercetin also exerts the suppressive effects on angiogenic factor production in vivo using experimental mouse model of allergic rhinitis. I revised the section of Materials and Methods (insert section 2.6. to 2.8.), Results (insert section 3.3. and 3.4.) and Discussion (insert lines 294 to 308), which are indicated with underline. I created new 2 figures (Figure 3 and Figure 4) by using new data.  2) The molecular mechanisms of inhibitory action of quercetin on angiogenic factor production were speculated in the last paragraph of discussion section (Lines 309 – 317) by using new 2 references (no.40 and No. 41). 3) I revised the introduction section (Lines 81 – 93).

Round 2

Reviewer 1 Report

1. The literature is also improperly cited.
ex:line 54, ref 9,10. IL-6's Ref? bFGF's Ref?
2. In the introduction or discussion, the role of angiogenic factor (ex: bFGF, IL-6) produced by mast cells in the "tissue remodeling" of allergic diseases is not described in detail. It was briefly mentioned only in ref 12, 13 (previously published by the laboratory), but these refs also did not mention it. Therefore, it cannot be seen that the production of angiogenic factors by mast cells plays an important role in tissue remodeling. Therefore, it is strange that this study used mast cells as the object in the in vitro experiment.
3. In addition, it is known that mast cells produce many kinds of inflammatory substances or interleukins, which is not innovative.

Author Response

I greatly appreciate your valuable comments. My replies to your specific comments are as follows. Revised portions are marked with underlines.

Comment 1, 2 & 3: According to these comments, I revised the introduction section (Lines 57 to 69). To do this, I inserted new 6 references (No. 10 to 15).

Other changes:

According to the comment raised by Referee 2, I revised the section of Material and Methods, 2.7 Collection of nasal lavage fluids.

I also re-drew figures 3 and 4.

Reviewer 2 Report

The manuscript has improved significantly by adding the experimental data on the in vivo effects of quercetin in an allergic mouse model. I have a few remaining issues which need the attention of the authors:

  1. Line 145- The method of the nasal fluid collection should be described in more detail in this manuscript. The authors now refer to an article which refers to another article etc, which is very annoying for the reader.
  2. In the allergic mouse model, the inhibitory effects of quercetin show a very narrow dose dependency. At 20 mg/kg no effect is seen and at 25 mg/kg one reaches maximal inhibition. Is it right that 30 mg/kg does not reach statistical significant inhibition? How does the in vivo concentrations reached at these dosages relate to the in vitro concentrations needed to inhibit mast cell activation. How do the authors explain these findings?
  3. In legends of figure 3 and 4 it should be explained what asterisks indicates.
  4. Line 318. Authors state: "The minimum dose that caused significant inhibition of these parameters was 25.0 mg/kg, which is similar to recommended dosage of quercetin for health maintenance
    [41]." This information cannot be found in reference 41. 

Author Response

I greatly appreciate your valuable comments. My replies to your specific comments are as follows. Revised portions are marked with underlines.

Comment 1: According to this comment, I revised the materials and methods section (Lines 146-151). To do this, new 9 references were added (No. 24 - 32).

Comment 2: This was stated in the discussion section (Lines 266-271). In Japan, the dosage of drug is determined by weighing 50kg to 55kg. Therefore, when converted to 25 mg/kg, it is 1250 mg, which is comparable to the maximum intake concentration shown in the literature.

Comment 3: According to this comment, I revised figures 3 and 4.

Comment 4: Thank you for your finding my mistake. I changed the reference.

Other Changes:

According to the comments raised by Referee 1, I revised the introduction section (Lines 57 to 69). To do this, I inserted new 6 references (No. 10 to 15).

Round 3

Reviewer 1 Report

There is a severe problem of self-plagiarism in the figure legends and material and methods. The author should rewrite the content of materials and methods and figure legends instead of plagiarizing the sentence of the previously published article.

Author Response

Referee 1

I greatly appreciate your valuable comments. My replies to your specific comments are as follows. Revised portions are marked with “Track Changes” function in Microsoft Word.

Comment 1: According to your comment, I revised the figure legends and the section of Materials and Methods.

Other changes: I changed title of the manuscript.

Reviewer 2 Report

The revised manuscript is acceptable for publication

Author Response

Referee 2

I am very happy to read your comment showing the acceptability of my manuscript. However, according to comment raised by Referee 1, I revised the manuscript.  Revised portions are marked using “Track Changes” function in Microsoft Word.

Changes: According to the comments raised by Referee 1, I revised the figure legends and the section of Materials and Methods.

Other changes: I changed title of the manuscript.